# Consequences of Acute or Chronic Methylphenidate Exposure Using Ex Vivo Neurochemistry and In Vivo Electrophysiology in the Prefrontal Cortex and Striatum of Rats [note 1]

**DOI:** 10.3390/ijms23158588

**Published:** 2022-08-02

**Authors:** Mathieu Di Miceli, Asma Derf, Benjamin Gronier

**Affiliations:** 1Pharmacology and Neuroscience Research Group, Leicester School of Pharmacy, De Montfort University, The Gateway, Leicester LE1 9BH, UK; dr.asmaderf@gmail.com (A.D.); bgronier@dmu.ac.uk (B.G.); 2Worcester Biomedical Research Group, School of Science and the Environment, University of Worcester, Worcester WR2 6AJ, UK

**Keywords:** ADHD, methylphenidate, neurotransmitter release, striatum, medium spiny neurons, pyramidal neurons

## Abstract

Methylphenidate (MPH) is among the main drugs prescribed to treat patients with attention-deficit and hyperactivity disease (ADHD). MPH blocks both the norepinephrine and dopamine reuptake transporters (NET and DAT, respectively). Our study was aimed at further understanding the mechanisms by which MPH could modulate neurotransmitter efflux, using ex vivo radiolabelled neurotransmitter assays isolated from rats. Here, we observed significant dopamine and norepinephrine efflux from the prefrontal cortex (PFC) after MPH (100 µM) exposure. Efflux was mediated by both dopamine and norepinephrine terminals. In the striatum, MPH (100 µM) triggered dopamine efflux through both sodium- and vesicular-dependent mechanisms. Chronic MPH exposure (4 mg/kg/day/animal, voluntary oral intake) for 15 days, followed by a 28-day washout period, increased the firing rate of PFC pyramidal neurons, assessed by in vivo extracellular single-cell electrophysiological recordings, without altering the responses to locally applied NMDA, via micro-iontophoresis. Furthermore, chronic MPH treatment resulted in decreased efficiency of extracellular dopamine to modulate NMDA-induced firing activities of medium spiny neurons in the striatum, together with lower MPH-induced (100 µM) dopamine outflow, suggesting desensitization to both dopamine and MPH in striatal regions. These results indicate that MPH can modulate neurotransmitter efflux in brain regions enriched with dopamine and/or norepinephrine terminals. Further, long-lasting alterations of striatal and prefrontal neurotransmission were observed, even after extensive washout periods. Further studies will be needed to understand the clinical implications of these findings.

## 1. Introduction

The mechanism by which attention-deficit and hyperactivity disease (ADHD) drugs exert their therapeutic effects, particularly on attention and cognition processes, remains to be fully elucidated. The efficacy of ADHD drugs resides in the ability to dampen hyperactivity while also improving cognition [1,2]. Although apparently safe to use, ADHD drugs require adequate dosing to avoid side effects [3,4,5]. Many drugs are available to treat ADHD, such as methylphenidate (MPH), D-amphetamine (D-amph), atomoxetine (ATX), bupropion, clonidine and reboxetine, although not all have received approval from the Food and Drug Administration [2]. Both MPH and D-amph have immediate effects on ADHD symptoms, whereas ATX has a longer onset of action, usually between 4 and 8 weeks [6,7].

D-amphetamine and MPH are strong inhibitors of the synaptic reuptake of both dopamine and norepinephrine. MPH potently inhibits the dopamine reuptake transporter (Ki = 34 nM) as well as the norepinephrine reuptake transporter (K_i_ = 339 nM) [8], although these values are likely underestimated and/or biased [9]. Other effects of D-amph include the inhibition of monoamine oxidase and blockade of vesicular transport of catecholamines [4,10]. On the other hand, ATX interacts very selectively with the norepinephrine transporter [8]. It is believed that the therapeutic effects of these drugs are associated with their abilities to stimulate dopamine release in the prefrontal cortex (PFC) [11,12], which is considered as among the main brain regions involved in the behavioral-calming and cognition-enhancing effects of ADHD drugs [13]. It also plays a critical role in the control of higher cognitive function such as vigilance, attention, impulsivity and behavioral inhibition [14]. According to microdialysis studies, MPH, at therapeutic doses (1–3 mg/kg), increase dopamine release preferentially in prefrontal areas, with little or no effect in basal ganglia [15]. Prefrontal dopamine, at the adequate concentration range, is thought to play a major positive role in cognition, attention and working memory, mainly through stimulation of dopamine D1 receptors [16]. Nevertheless, dopamine innervations are relatively sparse in the PFC [17]. Because dopamine has an affinity for the norepinephrine transporter (NET) [18], it is believed that a significant part of the dopamine released in prefrontal areas is cleared by (or even originates from) norepinephrine terminals. Moreover, dopamine and norepinephrine can be simultaneously co-released in specific noradrenergic terminals [19], while dopamine can even be re-uptaken by the NET [18]. A recent study has elegantly demonstrated that both DAT and NET can be targeted by chronic MPH administration in rats, as well as vesicular monoamine transporter 2 (VMAT2) and dopamine D1 receptors [20], showing that long-term exposure to MPH can alter catecholamine functions, especially in developing brains [21].

In contrast, in the striatum, where dopamine innervations are dense [22], increased striatal dopamine transporters and low striatal activity have both been observed in adult patients with ADHD [23,24], two effects alleviated by MPH [25], although this is not observed in all patients [26]. A recent review has summarized the effect of MPH on cognitive functions in patients with ADHD [27]. Patients with ADHD present altered cortico-striatal functional connectivity [28], a characteristic that was also observed in rodents following early postnatal dopamine lesions [29]. Altered cortico-striatal connectivity has also been observed in a mouse model of Parkinson’s disease, when dopamine innervation in the dorsal striatum is low [30].

In the present study, we aim to characterize how acute MPH could alter neurotransmitter release in dopaminergic or adrenergic terminals using ex vivo neurotransmitter release experiments performed on both prefrontal cortex and striatum samples. Furthermore, we also investigated the impact of chronic MPH treatment on neurotransmitter release in the striatum as well as the electrical activities of both prefrontal pyramidal neurons and striatal GABAergic medium spiny neurons.

## 2. Results

### 2.1. MPH Induces Dopamine and Norepinephrine Efflux in the Prefrontal Cortex

In the prefrontal cortex (PFC), application of methylphenidate (MPH) at 100 µM (Figure 1A) triggered significant ex vivo dopamine release (Bonferroni post hoc test after significant two-way ANOVA, Appendix A). This effect was dependent on norepinephrine terminals, since incubation of radiolabelled dopamine (35 nM) in the presence of desipramine significantly dampened dopamine efflux induced by 100 µM of MPH (Figure 1A, Bonferroni post hoc test after significant two-way ANOVA). This was furthermore confirmed by assessing radiolabelled norepinephrine (67–83 nM) efflux following MPH exposure in PFC samples. Indeed, MPH, at 100 µM, induced significant norepinephrine efflux in the PFC (Figure 1B, Tukey’s post hoc test after significant one-way ANOVA). A lower dose of 10 µM of MPH did not induce dopamine efflux under any conditions. These results indicate that MPH can induce both dopamine and norepinephrine efflux in the PFC, an effect that is arising from both dopamine and norepinephrine terminals when applied at 100 µM.

### 2.2. Chronic MPH Increases the Firing Activity of PFC Pyramidal Neurons without Altering Glutamatergic Neurotransmission

We have demonstrated in a previous study that acute exposure to MPH increases the firing activity of PFC pyramidal neurons and potentiates NMDA-induced neurotransmission [31]. Since acute exposure to MPH can induce dopamine or norepinephrine efflux within the PFC (Figure 1), we examined if chronic MPH exposure could modulate PFC neurotransmission. In animals chronically treated with MPH (voluntary oral intake of 4 mg/kg/day, in two separate doses, dissolved in 10% *v/v* sucrose solution, followed by a washout period of 28 days, [32]), a significant long-term increase in firing activity of PFC pyramidal neurons was observed (Figure 2A), compared to control animals (10% *v*/*v* sucrose vehicle, 4 mL/kg/day in two separate doses; Mann–Whitney test). However, compared to control animals, no difference was found concerning the bursting activities of these neurons (Figure 2B, Mann–Whitney test) nor the total number of spontaneously discharging neurons (Figure 2C, unpaired *t*-test). Moreover, chronic MPH exposure did not alter glutamatergic neurotransmission in these neurons, since iontophoretic NMDA induced similar firing activity enhancements in control and chronically treated animals (Figure 2D, Appendix A). A recording example time-course of such an experiment is illustrated in Figure 2E, where local iontophoretic NMDA induce reversible dose-dependent potentiation of firing activities. Thus, these results suggest that chronic MPH can modulate the excitability of cortical pyramidal neurons, without altering the responses of such neurons to locally applied NMDA.

### 2.3. Dopamine Efflux, but Not Dopamine Release, Is Modulated by MPH in the Striatum

Here, we used different ex vivo experimental conditions to determine the different mechanisms involved (Figure 3A). First, decreasing sodium concentration in the superfusate from 125 mEq to 20 mEq (by isotonic substitution with choline chloride, Figure 3B) significantly reduced MPH-induced dopamine efflux (Figure 3A, Appendix A), suggesting that electrochemical gradients are necessary for MPH-induced dopamine efflux. When vesicular content was depleted using a 20-min period pre-incubation in the presence of 1 µM of reserpine (Figure 3B, Appendix A), dopamine efflux induced by 100 µM of MPH was significantly reduced (Figure 3A, Appendix A), suggesting that MPH requires catecholamine vesicular integrity. The sodium substitution used in this study is known to induce rapid membrane hyperpolarization [33], which should decrease dopamine efflux. Our results are compatible with the sodium dependency of dopamine reuptake. Indeed, under extracellular sodium depletion conditions, dopamine cannot be re-uptaken by the DAT, as sodium gradients are known to be the driving forces of dopamine transport [34]. In fact, a reduction in the sodium driving force induced, in itself, significantly dampened dopamine efflux (Figure 3B). Further, to confirm that 1 µM of reserpine is sufficient to induce vesicular content depletion (Figure 3A), we confirmed on a few samples that application of reserpine at 1 µM can induce successful dopamine depletion, assessed by significant striatal dopamine efflux (Figure 3C), reflecting vesicular emptying. Increasing reserpine concentration to 10 µM dose dependently increased this effect (Figure 3C). In comparison to the above results, the selective dopamine reuptake transporter inhibitor GBR-12909 also induced dopamine efflux from striatal samples (Figure 3D, Appendix A), but to lower levels than previously observed (Figure 3A).

To further confirm such results, we measured dopamine efflux from striatal samples while superfusing a buffer rich in KCl (20 mM), which is known to induce neuronal membrane depolarization [35], thus producing dopamine release (arising from vesicular fusion). Raising KCl concentrations from 2.5 to 20 mM produced temporary dopamine releases (Figure 3E). When MPH (100 µM) was applied concomitantly to KCl, only non-significant dopamine release was observed (Figure 3E,F, *p* = 0.07, Mann–Whitney test). Similar results were found following data resampling with 1000 replicates (not shown), validating our previous results. Altogether, these results suggest that MPH, in the striatum, induces dopamine efflux by reversing the dopamine reuptake transporter.

### 2.4. Tolerance to MPH and Dopamine in Chronically Treated Animals

When MPH was superfused ex vivo in striatal tissue from animals chronically treated with MPH (4 mg/kg/day, please see [36,37,38] for related pharmacokinetics), it induced significantly lower dopamine efflux than in control animals (Figure 4), suggesting tolerance to MPH by the DAT (Bonferroni post hoc test after significant two-way ANOVA, Appendix A).

To confirm these data, we recorded a small sample of GABAergic medium spiny neurons (MSN) in control and MPH-treated animals. In the striatum, MSN are mostly silent during in vivo extracellular recordings [39], due to their low resting membrane potentials [40]. We observed that iontophoretically applied dopamine could dampen or potentiate NMDA-induced firing activities of MSN, likely reflecting direct and indirect striatal pathways [41,42,43]. Following chronic MPH treatment, the efficacy of dopamine in modulating in vivo NMDA-induced firing activities was significantly altered (not shown). These preliminary data suggest that plastic mechanisms can occur in MSN following chronic MPH treatment.

Altogether, these results suggest that acute MPH strongly modulates neurotransmission and that chronic exposure to MPH could exert long-term effects, as observed before in another region of the basal ganglia [37].

Finally, to compare the results acquired with MPH, we also investigated the potential of ATX, another drug approved for ADHD treatment, to modulate ex vivo dopamine and norepinephrine efflux. In the PFC, 100 µM of ATX produced significant dopamine efflux (Figure 5A). This was significantly reduced when slices were pretreated with 10 µM of desipramine, suggesting that this effect could originate in adrenergic terminals. To confirm this, we measured norepinephrine efflux in the PFC and observed a significant norepinephrine efflux from PFC slices when 100 µM of ATX is applied (Figure 5B), confirming our previous results. In the striatum, 100 µM of ATX induced large dopamine efflux, an effect that was strongly prevented by superfusing a low-Na^+^ buffer or by reserpine (1 µM) pre-treatment (Figure 5C), suggesting that ATX might modulate dopamine vesicular release in the striatum.

## 3. Discussion

We have demonstrated that MPH, at 100 µM, could induce ^3^H-dopamine efflux in the PFC (Figure 1). This effect was partly due to dopamine efflux arising from norepinephrine terminals, as desipramine dampened MPH-induced dopamine release. This was confirmed by observing norepinephrine efflux after superfusion with MPH (Figure 1). These results suggest an involvement of the NET in MPH-dependent dopamine efflux, at least in the PFC. Our data support previous assumptions that in the PFC, MPH elicits dopamine efflux mainly via an inhibition of the NET, suggesting that extracellular dopamine in the PFC originates not only from dopaminergic terminals but also from noradrenergic ones, where dopamine can act both as a precursor for norepinephrine and as a co-transmitter [19,44]. Some studies have suggested that both dopamine and norepinephrine are located within the same dense core vesicles in noradrenergic terminals [45]. Ex vivo, the IC_50_ for epinephrine uptake inhibition by MPH was 0.85 µM in the PFC and 0.12 µM in the striatum [46]. In vitro, MPH has a similar K_i_ for the human NET (0.1 µM) and DAT (0.06 µM) [47]. This was also observed for the mouse NET (0.17 µM) and DAT (0.26 µM) [47]. Furthermore, when DAT levels are low, dopamine can be uptaken by the NET, at least ex vivo [18]. Previous microdialysis and neurochemical studies on DAT knockout mice, as well as on naive rats, have shown that selective NET inhibitors increase prefrontal dopamine efflux [8,18]. We remain unable to evaluate the exact contribution of the DAT in the effects of MPH to induce dopamine efflux in the PFC in our experimental conditions. The DAT probably contributes to increasing dopamine efflux as MPH can still exert its effects when the slices were previously loaded in the presence of desipramine. However, following the loading of the slices, redistribution of tritiated dopamine within both dopamine and norepinephrine terminals could have occurred.

The efficacy of MPH in inducing dopamine efflux in the PFC under our conditions is lower than under in vivo conditions using microdialysis techniques. Systemic administration of low doses of MPH (1–2 mg/kg), probably reaching a concentration in the low micromolar range near the catecholamine synapse [48,49], could stimulate dopamine efflux by more than 300% [8], although this could be explained by a detection bias, as reuptake blockade will artificially drive high neurotransmitter release detection. Thus, comparing ex vivo and in vivo results could be challenging. When administered on the intact brain, these drugs are likely to activate other neuronal circuitries, which will further potentiate the release of dopamine in the PFC. Such activation may occur at local levels, as applications of MPH by reverse microdialysis in the PFC or in the nucleus accumbens can still produce consistent large increases in dopamine efflux in vivo [50,51].

In the striatum, however, noradrenergic innervations as well as norepinephrine reuptake transporter levels are low [18,52,53], which is in line with previous published observation, showing that, in the striatum, the NET is responsible for dopamine uptake only when DAT levels reach critically low levels, as observed in Parkinson’s disease [54,55]. In the present study, we have shown that MPH triggers dopamine efflux only through sodium-dependent mechanisms (Figure 3). The fact that pre-incubation with reserpine partially affected (*p* = 0.08) the ability of MPH to induce striatal dopamine efflux (Figure 3) suggests that inhibition of the DAT by MPH might involve vesicular integrity, at least to some extent. In our experimental design of sodium depletion, no osmolarity shock could have occurred due to the isotonic addition of choline chloride, thus preventing astrocyte swelling [56]. This sodium substitution is known to induce rapid membrane hyperpolarization [33], which should decrease dopamine efflux. Our results are compatible with the sodium dependency of dopamine reuptake [34]. Indeed, under extracellular sodium depletion conditions, dopamine cannot be re-uptaken by the DAT [57], as the sodium gradient is known to be the driving force of dopamine transport.

Our previous study has also demonstrated that MPH increases the firing discharges of PFC pyramidal neurons and can modulate NMDA neurotransmission in the PFC [31]. In the present study, chronic MPH induced long-lasting increases in the firing rates of PFC pyramidal neurons (Figure 2), an effect that was not linked to altered NMDA neurotransmission (Figure 2). In the striatum, chronic MPH exposure induced long-lasting desensitization to an acute 100 µM dose of MPH (Figure 4) or extracellularly applied dopamine (preliminary data not shown). These results are in line with one study witnessing either behavioral sensitization or tolerance (in a 1:1 ratio) to chronic MPH exposure (0.6–10 mg/kg for 5 days), correlated with electrophysiological sensitization or tolerance in MSN of the nucleus accumbens [58] and in PFC pyramidal neurons [59]. Interestingly, a recent study has pointed out that re-exposure to MPH can trigger behavioral sensitization or tolerance in rats, an effect not linked to the electrophysiological properties of dopaminergic neurons in the ventral tegmental area [60]. Further, adolescent or adult animals presented different responses to MPH [60], highlighting the importance of the exposure window. Chronic methylphenidate exposure may reduce striatal plasticity and might not be without long-term consequences, even after washout periods. In fact, chronic MPH exposure significantly decreased dopamine D2 receptor availability in the striatum [61,62], in line with our previous study showing dopamine D2 receptor desensitization in the midbrain following chronic MPH exposure during adolescence [32]. Altogether, these result may suggest long-term consequences of MPH on dopamine neurotransmission. Additional studies have also witnessed long-term consequence of MPH on brain metabolism [63], inflammation [64] and peripheral hemostasis [65]. Our results need to be studied further in order to determine the role of D1- or D2-like receptors after chronic exposure to MPH.

Previous studies have demonstrated that chronic MPH disrupts DA function in the striatum and synaptic plasticity, an effect that was dependent on dopamine D2 receptors [66]. Our previous study also observed similar results [32]. Chronic MPH is known to induce long-term consequences in the prefrontal cortex and striatum, such as sensitization/tolerance [59,67,68,69,70,71,72]. In the prefrontal cortex, these effects appear to be age-dependent [73] and can last for more than 10 weeks [74], which may warrant long-term monitoring in patients withdrawing from MPH. Furthermore, MPH is known to disrupt the expression of DAT and NET following chronic (but not acute) MPH exposure. Indeed, chronic MPH exposure increases DAT and NET levels in both striatal and prefrontal regions [20], as recently reviewed [75]. These results can explain, at least in part, the results observed in the present study. Furthermore, imaging studies in humans [76,77,78,79,80,81] have also observed significant effects of MPH on DAT and NET availability. In patients with ADHD, MPH treatment decreases DAT availability [82,83,84,85], likely indicating altered baseline DAT levels in these patients [24,83], although this is not always observed [81]. Finally, long-term MPH was shown to alter dopamine levels in striatal regions of patients with ADHD [86], highlighting altered dopamine neurotransmission following chronic MPH exposure.

In the current study, serotonergic terminals have not been examined. Since striatal dopamine terminals are known to co-release dopamine and serotonin [87] and that dopamine release in the striatum is dependent on serotonergic terminals [88], assessing striatal serotonergic-mediated release of dopamine following MPH exposure would be of interest. Due to the low concentrations of tritiated dopamine (35 nM) and norepinephrine (67–83 nM) used, recruitment of serotonergic terminals could not be achieved in the present study [89]. Thus, we were unable to determine if serotonergic terminals could be involved in dopamine and/or norepinephrine efflux following MPH exposure.

The present study also observed DA efflux following ATX exposure, an effect that was stronger than previously observed with MPH. Since ATX has a non-negligible affinity for the dopamine vesicle transporter (Ki ~ 3.5 µM) compared to MPH (Ki ~ 39.3 µM) [90], ATX could be uptaken inside the dopaminergic terminals through a relatively low affinity transporter, likely the DAT, where it will then interact with vesicular transporters (VMAT_2_, vesicular monoamine transporter 2) to promote intracellular dopamine efflux. Such dopamine efflux, observed at high ATX concentrations, may result from reverse dopamine transporter by the DAT as well as passive diffusion directly across phospholipid bilayers, which has been previously observed following amphetamine exposure [91].

To conclude, we have shown that acute exposure to the two ADHD drugs MPH or ATX can modulate dopamine and norepinephrine efflux in cortical and striatal structures. While norepinephrine terminals are likely the preferential targets within the PFC, dopamine terminals within the striatum can trigger dopamine outflows through sodium- and vesicular-dependent mechanisms. Chronic exposure to MPH induced desensitization of medium spiny neurons to locally applied NMDA, dampened striatal dopamine outflows during MPH exposure, increased firing rates of PFC pyramidal neurons in a long-lasting manner and desensitization to both MPH and dopamine in striatal regions. Further studies need to be conducted to evaluate the clinical implications of the current findings.

## 4. Materials and Methods

### 4.1. Animals and Drug Treatments

All animal experiments were conducted in strict accordance with the UK Home Office guidelines and the Animal Scientific Procedures Act (1986). A total of 27 male Sprague Dawley rats were housed in groups of 2–4 per cage, maintained at 20–22 °C with humidity rates above 40% under a 12:12 light/dark cycle with lights on at 07:00. Food and water were both provided ad libitum. Animals were allowed a three-day acclimatization period after delivery. All experiments were performed during the light phase and with permission from the UK Home Office (60/4333) and De Montfort University Ethics Committee. No adverse effects were reported.

In the present study, we used either naïve animals or late adolescent animals (150–180 g, PND 42) which were orally (per os) administered twice a day a sucrose solution (10% *v*/*v*, 2 mL/kg) with or without MPH (2 mg/kg), for 15 consecutive days, followed by a 28-day washout period, as detailed previously [32]. Daily MPH dose was 4 mg/kg/day/animal; given into two separate per os doses, each of 2 mg/kg. This protocol was chosen to best mimic peak plasma levels observed in patients treated with therapeutic MPH [36,37,49,92,93]. Furthermore, this regimen has also been used in our previous study [32]. Drug administration was therefore voluntary and stress free, which was adapted from a previously published protocol [94]. A washout period of 28 days was allowed before any experiments (neurochemistry or electrophysiology). For details about the pharmacokinetics of MPH, please see [38]. All experiments were performed on adult animals.

### 4.2. Ex Vivo Radiolabelled Neurotransmitter Efflux

Animals were sacrificed by cervical dislocation. The brain was quickly dissected out and immersed into ice-cold oxygenated Krebs buffer (NaCl 125 mM, MgSO_4_ 1.2 mM, KCl 2.5 mM, CaCl_2_ 2.5 mM, KH_2_PO_4_ 1.2 mM, NaHCO_3_ 25 mM, glucose 10 mM and pargyline 10 μM, to inhibit monoamine catabolism, pH 7.4). The brain was then placed on an ice-cold platform for further dissection of either the prefrontal cortex or the striatum. The tissue was then sliced into 350 × 350 µm prisms using a McIlwain tissue chopper (Campden Instruments LTD, Loughborough, UK). Constant oxygenation was maintained after this step (95% O2, 5% CO2). Prisms were then left for 20 min to rest at room temperature. Tissue prisms were then loaded for 40 min at 37 °C with either radiolabelled ^3^H-dopamine (1.0 µCi/mL, specific activity 28.7 Ci/mmol, 35 nM) or ^3^H-norepinephrine (1.0 µCi/mL, specific activity 12–15 Ci/mmol, 67–83 nM; Perkin-Elmer, Waltham, MA, USA) in the presence or absence of 10 µM of desipramine (to prevent norepinephrine uptake if necessary). Once the loading completed, the prisms were then washed 3 times with fresh Krebs buffer before being divided into perfusion chambers. Throughout the experiment, all samples and superfusion buffers were maintained at 37 °C. An equilibrating period of 40 min was initiated by superfusion of the chambers with Krebs buffer at 0.6 mL/min.

In order to determine baseline outflow of dopamine, 3–4 samples were collected per chamber at 4 min intervals. Sample were collected into vials and each sample would hold 2.4 mL of perfusion liquid, to which scintillation liquid was added up to a total volume of 7 mL per vial. At the end of the experiment, all tissues were collected and dissolved with 1 mL of tissue solubilizer. Total release quantities of tritium (^3^H) were measured in a liquid scintillation counter (Hidex 300 SL), from which disintegrations per minute were extracted. A minimum of 3 animals were used per experimental condition, except in Figure 3B,C (1 animal in each; internal controls).

If necessary, the composition of the superfused Krebs buffer was altered. Low-Na^+^ Krebs buffer consisted of 20 mEq of NaCl (instead of 125 mEq, substituted by isotonic concentration of choline chloride). A depolarizing buffer was also tested by increasing KCl concentration from 2.5 to 20 mM. Superfusion of such a K^+^-rich buffer (substituted by decreasing NaCl from 125 to 107 mM) is known to induce sudden membrane depolarization and neurotransmitter release [35].

### 4.3. In Vivo Extracellular Single-Unit Electrophysiology

Animals were initially deeply anaesthetized with urethane (1.2–1.7 g/kg, intraperitoneal, with additional doses administered if necessary), secured to a stereotaxic frame and maintained at 36–37 °C with a heating pad. A catheter was inserted into the lateral tail vein to perform systemic drug administration. An incision was made across the top of the head and the edges of the skin drawn back to reveal the cranium. Bregma was identified and a hole was drilled through the bone at the coordinates of the prefrontal cortex (PFC) or the striatum, according to the atlas of Paxinos and Watson [95]. Electrodes were manufactured in house from borosilicate capillaries (1.5 mm, Harvard Apparatus Ltd., Waterbeach, UK), pulled on a PP-830 vertical electrode puller (Narishige, Tokyo, Japan) and filled by hand with an electrolyte solution of NaCl 147 mM. The tip of the electrode was broken down under a microscope to an external diameter of 1–1.5 μm. Typical electrode resistance was in the range of 4–8 MΩ. Single-unit recordings with iontophoresis drug application were made using five-barrel glass micropipettes (World Precision Instruments, Hitchin, UK). The central recording barrel was filled with NaCl 147 mM. The side barrels were filled with: N-methyl-D-aspartate (NMDA) 30 mM and NaCl 2 M for current balancing. Outputs from the electrode were sent to a Neurolog AC pre-amplifier and amplifier (Digitimer Ltd., Welwyn Garden City, UK). If necessary, signal amplification was manually adjusted to record whole neuronal action potential amplitudes. Signals were filtered and sent to an audio amplifier, a Tektronix 2201 digital storage oscilloscope and a 1401 interface connected to a computer running Spike 2 v5.21 (Cambridge Electronic Design Ltd., Cambridge, UK) for data capture and analysis. Descent of the electrode was carried out using a hydraulic micromanipulator (MO-103, Narishige, Tokyo, Japan).

Putative glutamatergic pyramidal neurons were identified according to previous electrophysiological criteria: a broad action potential (1 ms), with a biphasic or triphasic, large waveform, starting with a positive inflection, a relatively slow firing rate, typically between 1 and 50 spikes/10 s and an irregular firing pattern [96,97].

Putative GABAergic medium spiny neurons were identified according to previous electrophysiological criteria such as a very low level or absence of spontaneous activity [39], in combination with a long-lasting action potential waveform, usually above 1 ms [98]. These neurons were detected by microiontophoretic applications of NMDA.

A minimum of 3 animals were used per experimental condition.

### 4.4. Drugs and Reagents

All drugs were purchased from Sigma (Sigma-Aldrich, Gillingham, UK). For neurotransmitter release assays, drugs were dissolved into normal or modified Krebs buffers, as appropriate. For intravenous administration during in vivo extracellular electrophysiology, all drugs were dissolved into saline (NaCl 0.9% *w/v*). For micro-iontophoresis experiments, all drugs (except NaCl 2 M) were dissolved into NaCl 147 mM.

### 4.5. Resampling

Resampling of means was performed with the Visual Inference Tool (VIT), a component of iNZight [99,100], which is encoded in R [101].

### 4.6. Data Analysis

All data are expressed as the mean ± standard error of the mean (S.E.M.). Statistical analyses were performed using paired/unpaired Student’s *t*-tests, Mann–Whitney tests (non-normal distribution) or one/two-way analysis of variance (ANOVA), followed by appropriate post hoc Tukey’s (one-way ANOVA) or Bonferroni tests (two-way ANOVA). Normal distributions were assessed with Shapiro–Wilk tests. The significance threshold was set at *p* < 0.05 and n values refer to the number of samples used. All statistical results are presented in Appendix A and were performed with R [101].

Fractional efflux for each superfusate sample was calculated by dividing the amount of tritium in each sample by the total tritium left thereafter. The effect of a tested condition was assessed on at least 3 subsequent sample collections and averaged. Normalized efflux values are calculated for each chamber as the ratio between the mean tested values (generally from at least 3 collections) and average baseline values (usually 3–4 collections). In the present study, we make a distinction between dopamine release and dopamine efflux/outflow. Indeed, dopamine release arises from vesicular exocytosis under in vivo or artificially stimulated conditions (e.g., perfusions of KCl), while dopamine efflux/outflow occurs when samples are not under stimulated conditions.

For resampling, 1000 replicates (r = 1000) of the initial data were generated. Output means (n = 1000 means) were analyzed using the 95% confidence interval (CI) spread.

## Figures and Tables

**Figure 1 ijms-23-08588-f001:**
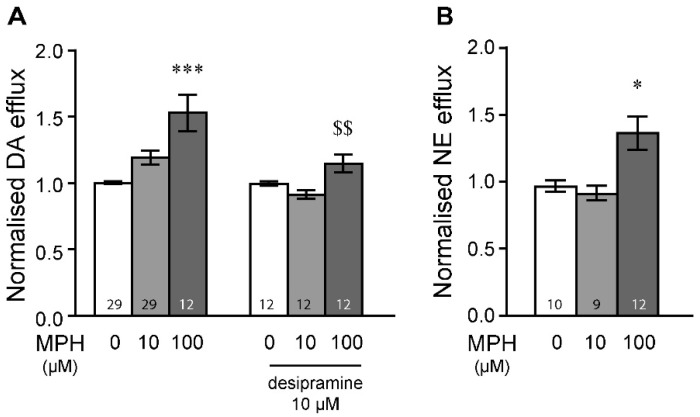
Methylphenidate induce dopamine and norepinephrine efflux in the prefrontal cortex. (**A**) Application of 100 µM of MPH, but not 10 µM, significantly induced dopamine efflux in the PFC, which was efficiently reduced when the norepinephrine transporter inhibitor desipramine (10 µM) was applied as a pre-treatment. (**B**) MPH at 100 µM induced significant norepinephrine efflux in the PFC. * *p* < 0.05 and *** *p* < 0.001 vs. respective 0 µM conditions; $$ *p* < 0.01 vs. 100 µM without desipramine. Bonferroni post hoc tests after significant two-way ANOVAs (**A**) or Tukey’s post hoc tests after significant one-way ANOVAs (**B**). n values are given for each condition and represent number of tissue samples used. DA: dopamine, NE: norepinephrine.

**Figure 2 ijms-23-08588-f002:**
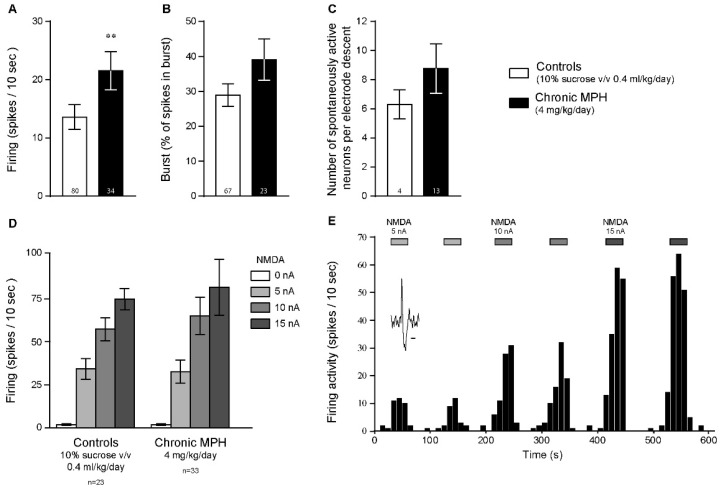
Chronic MPH treatment increases the firing activities of PFC pyramidal neurons, without altering NMDA neurotransmission. Chronic MPH (4 mg/kg/day/animal for 15 consecutive days followed by a 28-day washout period) significantly increased the firing discharges of PFC pyramidal neurons (**A**) but had no influence on the bursting discharges of these neurons (**B**) or the total number of spontaneously active neurons encountered during electrode descent (**C**). ** *p* < 0.01 vs. controls, unpaired *t*-tests. (**D**) Chronic MPH treatment did not influence the responses of PFC pyramidal neurons to locally applied NMDA (5–15 nA, 30 mM NMDA solution), via micro-iontophoresis (F_(3,162)_ = 0.26, *p* > 0.8). (**E**) Single-unit extracellular electrophysiological recording time course example, where NMDA increased the firing activities of a PFC pyramidal neuron in a current-dependent manner. A typical action potential waveform in such neurons is also illustrated. The horizontal bar represents 1 ms. n values are given for each group and represent number of neurons recorded, except in (**C**) where n values represent number of electrode descents.

**Figure 3 ijms-23-08588-f003:**
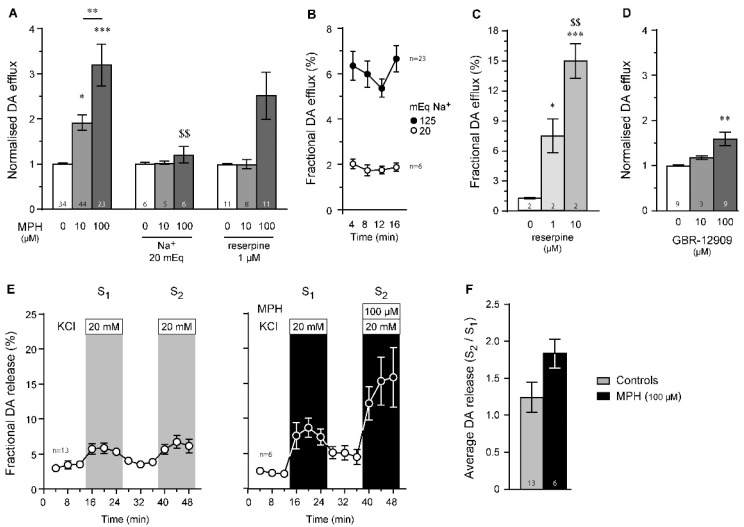
Effects of MPH on striatal dopamine efflux or dopamine release. (**A**) MPH-induced (10 and 100 µM) striatal dopamine efflux depended on sodium gradients and vesicular content. * *p* < 0.05, ** *p* < 0.01 and *** *p* < 0.001 vs. respective 0 µM conditions; $$ *p* < 0.01 vs. respective 100 µM conditions. Bonferroni post hoc tests after significant two-way ANOVAs. (**B**) Sodium depletion decreases dopamine efflux per se (Bonferroni post hoc test after significant two-way ANOVA (F_(1,27)_ = 17.67, *p* < 0.001). (**C**) The vesicle depleting agent reserpine successfully induces dopamine efflux in a dose-dependent manner. * *p* < 0.05 and *** *p* < 0.001 vs. 0 µM; $$ *p* < 0.01 vs. 1 µM. Tukey’s post hoc tests after significant one-way ANOVA. (**D**) In contrast, the selective DAT inhibitor GBR-12909 induced small but significant dopamine efflux when applied at 100 µM (** *p* < 0.01 vs. respective 0 µM, Tukey’s post hoc test after significant one-way ANOVA). (**E**) Dopamine release can be triggered by exposure to a KCl-rich Krebs buffer (20 mM instead of 2.5 mM). (**F**) However, MPH (100 µM) failed to significantly increase such dopamine releases (Mann–Whitney test, *p* = 0.07). n values are given for each condition and represent number of tissue samples used. DA: dopamine.

**Figure 4 ijms-23-08588-f004:**
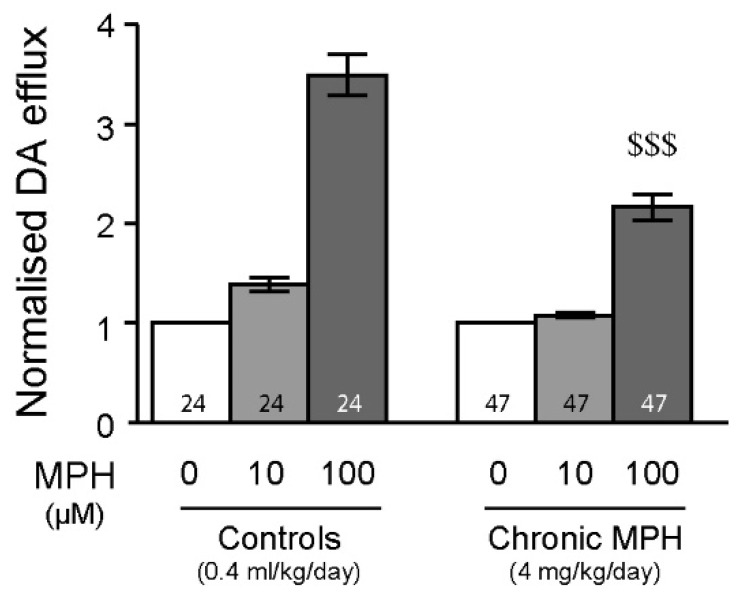
Chronic MPH treatment induces ex vivo long-term desensitization to subsequent MPH exposure in the striatum. Chronic treatment with MPH (4 mg/kg/day/animal for 15 days, followed by 28 days of washout) induced tolerance to subsequently applied MPH. Bonferroni post hoc tests after significant two-way ANOVAs, $$$ *p* < 0.001 vs. 100 µM in controls (two-way ANOVA results: [doses of MPH]: F_(2,138)_ = 276.4, *p* < 0.001; [chronic treatment]: F_(1,169)_ = 34.04, *p* < 0.001, [doses of MPH x chronic treatment]: F_(2,138)_ = 33.55, *p* < 0.001). n values are given for each condition and represent number of tissue samples used. DA: dopamine.

**Figure 5 ijms-23-08588-f005:**
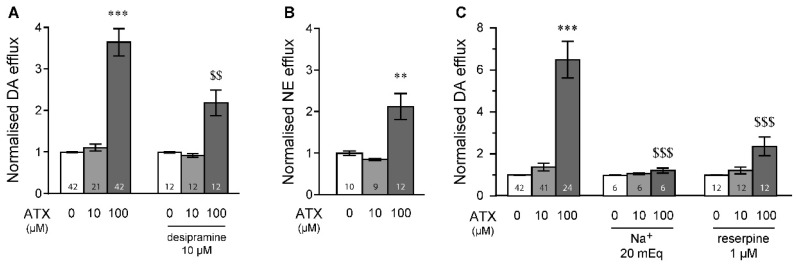
ATX modulates neurotransmitter efflux in the PFC and striatum. (**A**) Application of 100 µM of ATX significantly induced dopamine efflux in the PFC, which was significantly reduced when the norepinephrine transporter inhibitor desipramine (10 µM) was applied as a pre-treatment. Bonferroni post hoc tests after significant two-way ANOVAs. (**B**) ATX at 100 µM also induced significant norepinephrine efflux in the PFC. (**C**) Striatal dopamine efflux induced by ATX under baseline, low-Na^+^ buffer and pre-treatment with reserpine. n values are given for each condition and represent number of tissue samples used. Please refer to Appendix A for statistical results. *** *p* < 0.001, ** *p* < 0.01 vs. 0 µM. $$$ *p* < 0.001, $$ *p* < 0.01 vs. 100 µM in control.

## Data Availability

The data presented in this study are available on request from the corresponding author.

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
