# Peer review of "Consequences of Acute or Chronic Methylphenidate Exposure Using Ex Vivo Neurochemistry and In Vivo Electrophysiology in the Prefrontal Cortex and Striatum of Rats"

_ijms, 2022, doi:10.3390/ijms23158588_

Round 1

Reviewer 1 Report

The article by Di Miceli et al addresses the consequences of acute and chronic exposure of methylphenidate (MPH) on catecholaminergic (dopamine –DA - and noradrenaline -NA) transmission in the cortex and the striatum as well as the responsiveness of glutamatergic and gabaergic executive systems. The experiments have been performed in vitro (neurochemistry with preloaded 3H-neurotransmitter) or in vivo using single cell extracellular recordings with iontophoretic application of NMDA, after having exposed rats to MPH (chronic protocol). The context of the study being ADHD, the authors also addressed the reactivity of the system to atomoxetine, a preferential blocker of the norepinephrine transporter that is used in the treatment of ADHD as well. The authors nicely confirmed that 3H-DA efflux occurred from both adrenergic and dopaminergic terminals in the cortex. The situation in the striatum was different due to the high representation of dopaminergic terminals and the low concentration of adrenergic fibers. They also characterized the nature of the release of DA induced by MPH and atomoxetine using different approaches (low Na+, reserpine, high K+) to indicate that efflux of DA induced by MPH was sodium and vesicular dependent. After chronic exposure to MPH, MPH was less efficient to release DA in the striatum, increased basal firing rate of cortical neurons without altering their responsiveness to NMDA, and reduced the modulatory role of DA on NMDA-induced stimulation of striatal neurons. The authors conclude that MPH induced long-lasting alterations concerning the dopaminergic transmission along the corticostriatal tract.

The article is interesting and mixes neurochemistry and electrophysiology. The link between the two approaches is not always evident, but in the whole piece of work, it allows for highlighting specific changes on modulatory systems versus the executive systems. I have some remarks to ameliorate the article that is already well written.

1. The authors indirectly indicate the concentration for incubating 3H-DA and 3H-NA, which upon my calculation should be around 35 nM for DA and higher for NA. I would like the authors to indicate the actual values. And I would like also the authors to take into an account (maybe in the discussion) that serotonergic terminals would be not recruited when incubating the slices at these concentrations.

2. The authors paid attention to the osmolarity with the low Na+ concentration, but they did not mention any substitution for the high K+ medium, at least in the methods. Could thay add a comment on this?

3. Interpretation of the data, line 169: “Altogether, these results suggest that MPH, in the striatum, induces dopamine efflux solely by blocking the dopamine reuptake transporter. “ I do not understand the intent of the authors. The experiments reported in Figure 3 suggests that the effect of MPH relies on the reversal of the DAT, and not a simple blockade of the DAT.

4. ATX has strong effects on DA efflux in the striatum that are partially reduced by loss of Na+ of reserpine. Thus, it is slightly different from MPH, but I cannot rule out that the effects in the striatum are related to the blockade of the DAT. The authors could be aware that very high concentrations of these compounds will not guarantee specific blockade of the target (100 µM ATX is very high) and most of the uptake blockers behave as amphetamine when the concentrations are excessive. It has been described for some SSRI towards the SERT and in my hands and probably others, nomifensine (non exocytotic release). At least, the authors should discuss the role of the DAT in the effects of ATX.

5. Interpretation, line 256. The authors are indicating that the magnitude of the effect of MPH is much lower compared to the one that was published in vivo using microdialysis. Well, microdialysis is not the best technique to monitor extracellular levels of monoamines when the transporters are blocked. There is a big bias that is not considered but which is technically the following: by blocking the clearance of the monoamine, it favors the entry of the monoamine inside the probe giving the impression that the effects are massive. The best example is the release of dopamine produced by L-DOPA in dopamine neurons-depleted animals. The effects are massive, in part due to the absence of DAT, but the actual extracellular concentration of DA are in fact low. Thus, it was a good for the authors to discuss the existence of local interaction between cells, an argument that has to be present, but I would like the authors to slightly minimize the difference of magnitude between their approach and microdialysis data.

6.

Line 46: D-amph has not been mentioned as an abbreviation.

Line 120: iontophoretic (application of) NMDA?

Author Response

Reviewer 1

The article by Di Miceli et al addresses the consequences of acute and chronic exposure of methylphenidate (MPH) on catecholaminergic (dopamine –DA - and noradrenaline -NA) transmission in the cortex and the striatum as well as the responsiveness of glutamatergic and gabaergic executive systems. The experiments have been performed in vitro (neurochemistry with preloaded 3H-neurotransmitter) or in vivo using single cell extracellular recordings with iontophoretic application of NMDA, after having exposed rats to MPH (chronic protocol). The context of the study being ADHD, the authors also addressed the reactivity of the system to atomoxetine, a preferential blocker of the norepinephrine transporter that is used in the treatment of ADHD as well. The authors nicely confirmed that 3H-DA efflux occurred from both adrenergic and dopaminergic terminals in the cortex. The situation in the striatum was different due to the high representation of dopaminergic terminals and the low concentration of adrenergic fibers. They also characterized the nature of the release of DA induced by MPH and atomoxetine using different approaches (low Na+, reserpine, high K+) to indicate that efflux of DA induced by MPH was sodium and vesicular dependent. After chronic exposure to MPH, MPH was less efficient to release DA in the striatum, increased basal firing rate of cortical neurons without altering their responsiveness to NMDA, and reduced the modulatory role of DA on NMDA-induced stimulation of striatal neurons. The authors conclude that MPH induced long-lasting alterations concerning the dopaminergic transmission along the corticostriatal tract.

The article is interesting and mixes neurochemistry and electrophysiology. The link between the two approaches is not always evident, but in the whole piece of work, it allows for highlighting specific changes on modulatory systems versus the executive systems. I have some remarks to ameliorate the article that is already well written.

We would like to thank the reviewer for time spent on reviewing our work. We are delighted that the reviewer found the article of interest and for the positive appraisal of our work. Following these suggestions, our manuscript has been updated accordingly. Please refer to the yellow highlighting in the newest version of our manuscript for visual identification.

  1. The authors indirectly indicate the concentration for incubating 3H-DA and 3H-NA, which upon my calculation should be around 35 nM for DA and higher for NA. I would like the authors to indicate the actual values. And I would like also the authors to take into an account (maybe in the discussion) that serotonergic terminals would be not recruited when incubating the slices at these concentrations.

We thank the reviewer for pointing this out. The reviewer is correct: the concentration of 3H-dopamine is indeed 35 nM, and 67-83 nM for 3H-norepinephrine. This has been inserted in the method (lines 386 and 387) as well as the beginning of the results (lines 89 and 92). Furthermore, we have included the potential limitation of our study regarding not recruiting serotonergic terminals using these concentrations (please see the updated discussion, lines 327-334).

  1. The authors paid attention to the osmolarity with the low Na+ concentration, but they did not mention any substitution for the high K+ medium, at least in the methods. Could they add a comment on this?

We apologise for forgetting to mention this. The increase of 17.5 mM of K+ (from 2.5 to 20 mM) was alleviated by decreasing NaCl from 125 to 107 mM, to avoid creating osmotic chocks. This has now been updated on lines 404-405.

  1. Interpretation of the data, line 169: “Altogether, these results suggest that MPH, in the striatum, induces dopamine efflux solely by blocking the dopamine reuptake transporter. “ I do not understand the intent of the authors. The experiments reported in Figure 3 suggests that the effect of MPH relies on the reversal of the DAT, and not a simple blockade of the DAT.

The reviewer is right, Figure 3 indeed shows reversal of the DAT, not blockade. This was an error and is now corrected (now on line 174). We apologise for any confusion that was created.

  1. ATX has strong effects on DA efflux in the striatum that are partially reduced by loss of Na+ of reserpine. Thus, it is slightly different from MPH, but I cannot rule out that the effects in the striatum are related to the blockade of the DAT. The authors could be aware that very high concentrations of these compounds will not guarantee specific blockade of the target (100 µM ATX is very high) and most of the uptake blockers behave as amphetamine when the concentrations are excessive. It has been described for some SSRI towards the SERT and in my hands and probably others, nomifensine (non exocytotic release). At least, the authors should discuss the role of the DAT in the effects of ATX.

We thank the reviewing for his interesting suggestion. We have now added a new paragraph in our updated manuscript to reflect this. Please see lines 335-343, explaining such a possibility and the rationale behind it.

  1. Interpretation, line 256. The authors are indicating that the magnitude of the effect of MPH is much lower compared to the one that was published in vivo using microdialysis. Well, microdialysis is not the best technique to monitor extracellular levels of monoamines when the transporters are blocked. There is a big bias that is not considered but which is technically the following: by blocking the clearance of the monoamine, it favors the entry of the monoamine inside the probe giving the impression that the effects are massive. The best example is the release of dopamine produced by L-DOPA in dopamine neurons-depleted animals. The effects are massive, in part due to the absence of DAT, but the actual extracellular concentration of DA are in fact low. Thus, it was a good for the authors to discuss the existence of local interaction between cells, an argument that has to be present, but I would like the authors to slightly minimize the difference of magnitude between their approach and microdialysis data.

We thank the reviewer for pointing out this possible bias. We have dampened our approach by changing from “is much lower” towards “is lower”, on lines 261-262, and have also inserted a sentence to explain such a possible bias (lines 265-267).

6.

Line 46: D-amph has not been mentioned as an abbreviation.

This change has been made, apologies (please refer to line 39 in our updated manuscript).

Line 120: iontophoretic (application of) NMDA?

This change has been made, thank you. Please see line 125 of our updated manuscript.

Reviewer 2 Report

Response to authors

The insight of this study is fascinating, and the authors provided many references to ensure the dose they used was suitable. Although the MPH modulates DA and NE efflux have been reported, the authors provided more evidence to suggest that the methylphenidate (MPH) could modulate DA and NE efflux with DA or NE terminal in PFC and DA efflux in the striatum and also showed the chronic MPH exposure effects. The results are generally precise, but some concerns and questions need to be explained.

1.     Please provide the IACUC numbers. Furthermore, the sample size shown in figures (such as Figure1) indicates the used animal numbers or slice numbers?

2.     Which studies used naïve animals, and which studies used late adolescent animals? Any difference between these two groups of animals in the current study?

3.     Please check that all figure indication in the contents is correct; for example, Figure 3D is missing a label.

4.     In the current study, the authors showed that the firing rate increases in the PFC and long-term desensitization in DA efflux in the striatum after chronic MPH administration. Whether the firing rate alteration affects the DA efflux? How about the firing rate alteration after re-exposure to MPH in the PFC and striatum of chronic MPH exposure animals?

5.     Although the authors pointed out that D2 desensitization in the midbrain followed chronic MPH exposure and was tried to link to behavior by reference citation. But, the DAT or NET expression may affect the DA efflux, which may have a more direct link to the current study's findings; how about the changes of DAT and NET expression or binding affinity of MPH to DAT and NET in PFC and striatum of chronic MPH exposure animals?

6.     please make consistent the term used for ex vivo and in vitro. The two terms are staggered in content and seem to indicate the same thing or experiment.

7.     The DA efflux could be induced by MPH. However, chronic MPH with 28 days wash-out period still showed DA efflux reduced after MPH re-exposure, which may imply the tolerance occurred after long-term MPH exposure without a wash-out period. Could the authors have any suggestions? Have any differences in neurotransmitters efflux and firing rate of chronic MPH exposure with/without a long-term (28 days) wash-out period? Moreover, chronic MPH exposure may reduce striatal plasticity, have any studies showed that how long of wash-out period could recover the chronic MPH-caused effects?

Author Response

Reviewer 2

Response to authors

The insight of this study is fascinating, and the authors provided many references to ensure the dose they used was suitable. Although the MPH modulates DA and NE efflux have been reported, the authors provided more evidence to suggest that the methylphenidate (MPH) could modulate DA and NE efflux with DA or NE terminal in PFC and DA efflux in the striatum and also showed the chronic MPH exposure effects. The results are generally precise, but some concerns and questions need to be explained.

We would like to thank the reviewer for time spent on reviewing our work and for the positive comments made. We are delighted that the reviewer found the article of interest. Following these suggestions, our manuscript has been updated accordingly. Please refer to the yellow highlighting in the newest version of our manuscript for visual identification.

  1. Please provide the IACUC numbers. Furthermore, the sample size shown in figures (such as Figure1) indicates the used animal numbers or slice numbers?

We thank the reviewer for asking this information. No adverse effects were reported (now written on line 362). Furthermore, a total of 27 male rats were used (updated on line 356). Each figure legend details the number of samples used (whether neurons recorded or tissues preparations; highlighted in yellow). A minimum of 3 animals were used per experimental condition, except in two internal control experiments (Figure 3B and Figure 3C). This has now been inserted in our latest manuscript (please see lines 399-400 and line 437).

  1. Which studies used naïve animals, and which studies used late adolescent animals? Any difference between these two groups of animals in the current study?

Naïve adult animals were used for ex vivo neurotransmitter release experiments. Late adolescent animals were used at the beginning of the experiments involving chronic MPH exposure or vehicle. In vivo electrophysiological recordings and ex vivo neurotransmitter release experiments were all performed at adulthood for all animals, to avoid potential age-dependency of the observations, thus limiting potential bias. This is now explicitly written on line 373.

  1. Please check that all figure indication in the contents is correct; for example, Figure 3D is missing a label.

We apologise, the result narrative was not ordered indeed. In the updated version of the article, we have inserted the full indication for Figure 3D. Please see lines 162-165. We have checked the manuscript throughout, thank you for pointing this out.

  1. In the current study, the authors showed that the firing rate increases in the PFC and long-term desensitization in DA efflux in the striatum after chronic MPH administration. Whether the firing rate alteration affects the DA efflux? How about the firing rate alteration after re-exposure to MPH in the PFC and striatum of chronic MPH exposure animals?

Here, the reviewer is raising a very interesting question. These experiments were performed by the group of Nachum Dafny in several different papers. Our discussion now reflects these findings. Please see lines 311-318.

  1. Although the authors pointed out that D2 desensitization in the midbrain followed chronic MPH exposure and was tried to link to behavior by reference citation. But, the DAT or NET expression may affect the DA efflux, which may have a more direct link to the current study's findings; how about the changes of DAT and NET expression or binding affinity of MPH to DAT and NET in PFC and striatum of chronic MPH exposure animals?

The reviewer in raising interesting suggestions here. We have included a reference showing the impact of acute or chronic MPH exposure on the expression of the DAT and NET on lines 318-326.

  1. Please make consistent the term used for ex vivo and in vitro. The two terms are staggered in content and seem to indicate the same thing or experiment.

We apologise for creating confusion by using these two terms. “Ex vivo” has now been used throughout the manuscript, except when referring to a previously-published paper (ie: “in vitro”, line 250, referring to the 2006 article by Han and colleagues, which used transfected cells).

  1. The DA efflux could be induced by MPH. However, chronic MPH with 28 days wash-out period still showed DA efflux reduced after MPH re-exposure, which may imply the tolerance occurred after long-term MPH exposure without a wash-out period. Could the authors have any suggestions? Have any differences in neurotransmitters efflux and firing rate of chronic MPH exposure with/without a long-term (28 days) wash-out period? Moreover, chronic MPH exposure may reduce striatal plasticity, have any studies showed that how long of wash-out period could recover the chronic MPH-caused effects?

We thank the reviewer for these interesting comments. We have added a new paragraph at the end of our discussion to reflect these suggestions. In fact, the group led by Nachum Dafny has been focusing on these questions. Thus, we have included some of these studies to reflect how MPH has short- and long-term consequences (dopamine function, electrophysiology), can induce sensitization/tolerance, and how age can affect responses to MPH. Please see our updated manuscript on lines 311-318.

Round 2

Reviewer 2 Report

Response to authors

The authors have addressed all my concerns. The authors have provided more direct references, evidence, and discussions to improve their hypotheses and complete the story. I acknowledge the efforts put by the authors to integrate all the suggestions made by the reviewer. The manuscript could be considered for acceptance.